# Dispelling Myths about Antenatal TAPS: A Call for Action for Routine MCA-PSV Doppler Screening in the United States

**DOI:** 10.3390/jcm8070977

**Published:** 2019-07-04

**Authors:** Lauren Nicholas, Rebecca Fischbein, Julie Aultman, Stephanie Ernst-Milner

**Affiliations:** 1Department of Social Sciences, D’Youville College, 591 Niagara Street, Buffalo, NY 14201, USA; 2Department of Community and Family Medicine, Northeast Ohio Medical University, 4209 State Route 44, PO Box 95, Rootstown, OH 44272, USA; 3Twin Anaemia Polycythemia Sequence (TAPS) Support Group, TAPS Patient Advocate, 1326HS Almere, The Netherlands

**Keywords:** twin anemia-polycythemia sequence, TAPS, MCA-PSV Doppler, screening, clinical guidelines, monochorionic diamniotic twin pregnancy

## Abstract

In the United States, routine middle cerebral artery peak systolic velocity (MCA-PSV) Doppler screening for the detection of antenatal twin anemia-polycythemia sequence (TAPS) is not recommended. The current and only national clinical guideline from the highly-influential Society for Maternal-Fetal Medicine states that, “There is no evidence that monitoring for TAPS with MCA PSV Doppler at any time, including > 26 weeks, improves outcomes, so that this additional screening cannot be recommended at this time.” We argue this recommendation has disproportionate influence on patients and the care they are offered and receive. We use current evidence to highlight and dispel pervasive myths surrounding antenatal TAPS and the value of routine MCA-PSV screening. An ethical framework that illustrates the importance of giving patients the opportunity for routine screening is presented. Findings demonstrate that: (1) both spontaneous and post-laser TAPS is a serious, potentially life-threatening complication, (2) treatment for TAPS is effective and includes expectant management, intrauterine transfusion (IUT), or surgery, (3) and routine MCA-PSV, which has satisfactory diagnostic accuracy, is currently the only way to provide early detection of TAPS. We conclude that routine TAPS screening is a medically proven valuable resource that should be offered to patients in need and to the clinicians who are trying to act toward their benefit.

## 1. Introduction

Monochorionic (MC) twin pregnancies have long been fraught with a mortality rate that exceeds dichorionic twins by over seven times [1]. In addition, the incidence of congenital anomalies in monochorionic twinning is increased by > 2-fold over dichorionic twins [2] and 6 to 10-fold over singletons [3]. Inherent shared placental circulation sets the stage for a host of potentially life-threatening complications [4] and improved survival rates for this population can only be observed within our most recent human history, attributed nearly exclusively to the last 30 years of scientific innovation [5]. 

Twin anemia-polycythemia sequence was first described in 2006 by Robyr et al. [6]. The ‘spontaneously’ occurring form, as well as the acronym “TAPS,” was then described in 2007 by Lopriore et al. and marked a prominent and necessary advancement in the understanding and overall risk mitigation of MC multiples [7]. For those working to reduce morbidity and mortality within the MC population, post-2006 expansion of knowledge regarding TAPS etiology, diagnosis and staging criteria, and options for effective interventions, offer health care providers and their patients tools and choices that did not previously exist. 

However, many providers and clinical organizations have chosen not to embrace and utilize current-and-growing TAPS knowledge for reasons we discuss below.

Antenatal TAPS can only be treated if it is first diagnosed using middle cerebral artery peak systolic velocity (MCA-PSV) Doppler ultrasound [6,8,9,10]. When health care providers refuse to perform routine MCA-PSV screening for TAPS detection, all subsequent advancements and innovative technologies related to treatment are rendered useless, and shared decision making between provider and patient is impaired.

Such refusal can be seen in published clinical guidelines put forth by the Society for Maternal-Fetal Medicine (SMFM), wherein their recommendation states, “There is no evidence that monitoring for TAPS with MCA PSV Doppler at any time, including > 26 weeks, improves outcomes; so that this additional screening cannot be recommended at this time” [11]. Additionally, our own research regarding TAPS screening practices amongst Maternal-Fetal Medicine Specialists (MFMs) in the United States showed that over one-third of MFMs surveyed do not perform routine MCA-PSV screening [12]. Our research revealed that the reasons MFMs do not conduct routine MCA-PSV screening mirror the hesitation voiced by the SMFM statement: reasons were not due to lack of time or inability to obtain reimbursement for the screening, but rather the belief that MCA-PSV Doppler is not a reliable test, the belief that there is unclear treatment protocol for TAPS, the belief that there is only value in post-laser TAPS screening, and a lack of familiarity with TAPS monitoring methods [12]. The reasons provided by SMFM and MFMs to forgo MCA-PSV screening are not evidence-based, and we have thus chosen to refer to them as myths.

This paper seeks to use current information to dispel these myths while discussing inherent ethical issues borne from withholding the opportunity for routine MCA-PSV screening for MC patients. What we have provided is an argumentative review to refute deeply embedded assumptions and information, which have led to poorly examined guidelines and practices that negatively impact the welfare of our patient populations. The approaches to this paper follow the philosophical traditions of theoretical analysis, which is an accepted methodology in bioethics and the social sciences that intersect with medical research methods. The postulates, theoretical support, and argumentative framework substantiate our descriptive claims to achieve valid and sound logical argumentation to be considered for the advancement of future research and discussion in this area. 

### 1.1. MYTH 1: The Natural History of TAPS is Unknown and so MCA-PSV Screening Cannot be Recommended

TAPS, a complication arising from intertwin anastomoses connecting the fetal circulations, has been recognized, studied, and received considerable publication for over 12 years [6]. TAPS is caused by the slow transfer of blood from one fetus to the other via very few, small anastomoses [13].

This slow transfusion promotes anemia in one fetus and polycythemia in the other, while no serious difference occurs in the amounts of amniotic fluid [7]. Absence of twin oligo-polyhydramnios sequence (TOPS) is a condition sine qua non, since the presence of TOPS is pathognomonic for twin-twin transfusion syndrome (TTTS) [6,14,15]. TAPS represents a unidirectional feto-fetal transfusion of approximately 1% of the fetal blood volume, or approximately 5 to 10 mL, per 24 h [16]. This is in contrast to TTTS in which the anastomoses are often larger, bidirectional, and in which there is a net transfusion of approximately 2% of fetal blood volume per 24 h from donor to recipient. TTTS is the result of a combination of too high and too low blood volume. TAPS is about too high and too low blood concentration [17].

TAPS can occur spontaneously in 3% to 6% of otherwise uncomplicated MC twin pregnancies [14,18] or following laser ablation in up to 16% of cases [6]; it can occur even after the Solomon technique has been used [19]; though its incidence rate is much lower, following Solomon (3% versus 16% in standard treatment) [20]. Most antenatal TAPS cases are diagnosed in the second or third trimester [21]. It should be noted that TAPS remains an under-recognized and under-screened-for complication, and therefore, especially its spontaneous incidence, may be underestimated.

Antenatal TAPS is diagnosed based on measurement of the middle cerebral artery peak systolic velocity (MCA-PSV) [22]. The best way to use MCA-PSV to diagnose TAPS has evolved over time. Previous acceptable thresholds included MCA-PSV 1.5 multiples of the normal median (MoM) in one twin and below 0.8 MoM in the other twin [6], which was then replaced by MCA-PSV 1.5 MoM in one twin and below 1.0 MoM in the other twin [14]. However, Delta MCA-PSV > 0.5 MoM is now considered the best and most reliable reference standard for diagnosis of antenatal TAPS [8,23]. 

Like TTTS, antenatal TAPS is represented by stages with progression through stages culminating in fetal death. Also similar to TTTS, the progression of TAPS stages are not always linear, do not always progress, and may even regress [15]. The following staging represents the recently improved TAPS classification system by Tollenaar et al. [23], wherein TAPS stages include:

Stage 1: Delta MCA-PSV > 0.5 MoM, without signs of fetal compromise

Stage 2: Delta MCA-PSV > 0.7 MoM, without signs of fetal compromise

Stage 3: As Stage 1 or 2, with cardiac compromise of donor *

Stage 4: Hydrops of donor

Stage 5: Intrauterine demise of one or both fetuses preceded by TAPS

* Defined as critically abnormal flow: Doppler shows absent or reversed end-diastolic flow in umbilical artery, pulsatile flow in umbilical vein and/or increased pulsatility index or reversed flow in ductus venosus.

Other typical ultrasound TAPS findings include placental discordance; the anemic donor’s region of the placenta is thickened and hyperechoic, and the plethoric recipient’s region of the placenta appears normal [24] and starry sky liver in the recipient [25].

Given the rareness of MCDA pregnancies, much of the research regarding TAPS, including short and long-term outcomes, has been based on findings from case series and case studies [15]. Further, because TAPS is a heterogenous disorder, pre- and post-natal outcomes for TAPS can vary greatly, ranging from mild anemia and polycythemia to severe morbidity and mortality [9,15,26]. Specifically, severe anemia in the TAPS donor can cause fetal hydrops, severe acidemia, and perinatal death, while severe polycythemia in the recipient can cause impaired perfusion, skin necrosis, vascular limb occlusion, and prenatal death [27]. Neurodevelopmental impairment and severe cerebral injury can also occur [9,27,28]. The Leiden University Medical Center is currently running the TAPS trial wherein one objective is to track long-term outcomes of TAPS survivors. Early findings show that 9% of TAPS survivors experience long-term developmental impairment. While post-laser TAPS survivors have similar outcomes, spontaneous TAPS donors show increased risk of long-term impairment and hearing loss [29]. Additional findings are forthcoming. While further prospective research using larger samples is required to better understand the full natural history of the disorder, including pre- and post-natal TAPS outcomes, evidence remains that TAPS can be lethal and permanently life altering for TAPS survivors and their families. Importantly, without MCA-PSV screening, TAPS cannot be diagnosed or staged antenatally, effectively denying patients the opportunity for information and shared decision-making.

### 1.2. MYTH 2: MCA-PSV Doppler is not a Reliable Test for TAPS

Detection of antenatal TAPS is crucial, as it affects management of the pregnancy [18,30] and gives patients the information necessary to choose how they wish to proceed with their pregnancies. MCA-PSV Doppler measurement is a non-invasive test that has become the standard assessment for diagnosis of fetal anemia in singletons in a variety of fetal diseases with high diagnostic accuracy [31] and high diagnostic accuracy for moderate-severe anemia in transfused and untransfused singletons [32]. 

To obtain MCA-PSV measurement, an axial section of the brain, including thalami, cavum septi pellucidi and greater wing of sphenoid, with the circle of Willis identified by color Doppler, should be obtained. The MCA closest to the probe is sampled at or near its origin from the internal carotid artery. The waveform peak is measured, with the angle of insonation as close as possible to 0°, and always < 30°. Higher inter- and intra- observer variability results from angle correction and sampling of more distal regions of the MCA. The fetus should be quiescent, as heart-rate accelerations and movement can alter measurements [33]. A step-by-step video tutorial and an MoM calculator are available [29,34]. 

While MCA-PSV serves as a gold standard for diagnosis of anemia in singletons, the diagnostic accuracy and utility of MCA-PSV measurement for TAPS has been considered uncertain [35]. More specifically, the value of MCA-PSV in predicting polycythemia by itself is not as well established as its use for the prediction of fetal anemia. The National Institute for Health and Care Excellence (NICE), which establishes guidelines regarding healthcare in England, conducted a systematic review to determine the accuracy of prenatal MCA-PSV for predicting postnatal TAPS. Three studies met criteria for inclusion and, based on the potential for study bias with the included studies, NICE concluded that they could only recommend routine MCA-PSV screening when MC pregnancies were already complicated [35]. In Table 1, we report test accuracy from studies that used MCA-PSV MoM to diagnose TAPS, using criteria similar to that of NICE, specifically: studies must report test accuracy of MCA-PSV with n size of greater than 5. However, unlike the NICE review, we did not limit our reporting to those with a reference standard of postnatal TAPS and included an additional study which reported a reference standard of fetal anemia [36]. The four studies in Table 1 included in their protocols—a timely interval between antenatal MCA-PSV and delivery (no greater than 1 week), increasing confidence in the accuracy of the MCA-PSV measurements. Slaghekke et al. took 116 measurements from 43 twin pregnancies complicated by TAPS. The accuracy of MCA-PSV, measured immediately prior to fetal Hb measurement by fetal or cord blood sampling for prediction of anemia and polycythemia, was assessed [36]. MCA-PSV MoM values correlated well with Hb levels (r = −0.86, *p* < 0.001) and sensitivity and specificity of MCA-PSV ≥ 1.5 MoM to predict severe anemia in TAPS donors was 94% and 74%, respectively. The sensitivity of MCA-PSV ≤ 1.0 MoM to predict polycythemia in TAPS recipients was 97% and specificity was 96% [36]. The authors concluded that using a cut-off of > 0.8 MoM for severe polycythemia would result in too many missed cases.

A 2018 analysis of 45 uncomplicated MC and 35 TAPS twins by Tollenaar et al. showed that the sensitivity and specificity of the cut-off MCA-PSV values (> 1.5 MoM in the donor and < 1.0 MoM in the recipient) to predict postnatal TAPS was 46% and 100%, respectively [23]. Delta MCA-PSV > 0.5 MoM had a higher diagnostic accuracy for predicting TAPS and showed a sensitivity of 83% and a specificity of 100% [23]. Veujoz et al. report a sensitivity of 71% and specificity of 50%, however, their analyses were based on only nine cases [37]. Finally, findings from Fishel-Bartal et al. also demonstrate the utility of MCA-PSV for the risk assessment of TAPS [18] with a high AUC of 0.87. 

Findings in Table 1 suggest that delta MCA-PSV, rather than the actual MCA-PSV of each twin, is more accurate at detecting TAPS.

In addition to cut-off values, whether or not a fetus has been transfused can also affect the accuracy of MCA-PSV Doppler measurements for TAPS. A 2019 meta-analysis by Martinez-Portilla et al. examined observational studies evaluating the performance of MCA-PSV using a 1.5 MoM threshold for the prediction of fetal anemia [32]. The reference standard was fetal anemia by blood sampling. A total of 12 studies and 696 fetuses showed the area under the curve (AUC) for moderate-severe anemia was 83%. Pooled sensitivity and specificity were 79% and 73%, respectively. When only untransfused fetuses are considered, prediction improves, achieving an 87% AUC, 86% sensitivity, and 71% specificity. A decline in sensitivity is observed (estimate −0.055; 95% CI: −0.107 to −0.003; *p* = 0.039), as more transfusions are required [32]. 

Other factors that may influence MCA-PSV include: gender, cardiac status, uterine contractions, fetal behavioral state [38], and advanced gestational age [39]. Elevated MCA-PSV measurements may also be seen in abnormal placentation and IUGR, reflecting cerebral autoregulation in response to hypoxemia and hypercapnia [40]. These factors, in addition to insufficient attention to proper technique, may in part explain the false-positive rate of MCA-PSV for prediction of fetal anemia [41].

However, prediction may be improved through the use of serial MCA-PSV measurements to track trends (versus a single measurement), resulting in a reduction of the false-positive rate to less than 5% [42].

In summary, MCA-PSV Doppler is a non-invasive and accurate test for detecting fetal anemia, polycythemia, and TAPS. Sensitivity and specificity levels are high and acceptable. False positives and negatives can be reduced by adhering to the proper scanning technique, obtaining intra- and inter-rater consensuses, performing serial MCA-PSV readings, and remaining mindful of additional conditions that may alter readings. Finally, the recommended use of the improved antenatal classification system by Tollenaar et al. [23], which includes delta cutoffs, provides a more representative difference within the twins, maximizing the value of MCA-PSV Doppler screening.

### 1.3. MYTH 3: TAPS Presents with Other Symptoms that Will be Visible via Other Tests

Amongst the most systemic myths surrounding TAPS is that other abnormal ultrasound findings will always appear and/or that TAPS will always coexist with other fetal disorders that are detectable without MCA-PSV screening. It is this myth that inspires some health care providers to start screening for TAPS only when other complications such as TTTS or IUGR are presenting. While we applaud governing bodies such as NICE for recognizing and endorsing routine MCA-PSV screening to mitigate the dangers of TAPS to MC pregnancies already complicated by other syndromes [35], such recommendations leave patients with uncomplicated MC pregnancies vulnerable and with little recourse. 

The primary perspective about which abnormal findings “should be” observed alongside TAPS is discordant amniotic fluid levels. How this myth was developed, and why it still exists, is perplexing. The earliest publications describing TAPS and its natural history, and all of those since, are quick and clear to define TAPS as notable due to its non-presence of TOPS [43]; for this reason, routine MCA-PSV screening is urged. 

While ‘post-laser’ TAPS provides a very clear demarcation of when to begin MCA-PSV screening based on the premise of ‘other symptoms or complications,’ spontaneous TAPS is a notable threat, occurring at a rate of up to 6% in otherwise ‘uncomplicated’ MC pregnancies and will not always be accompanied by such qualifying symptoms [14]. 

It is important to note we are not arguing that other complications cannot accompany TAPS—they certainly can. The argument is that other complications or indications are not common nor required and TAPS screening should not be predicated upon their development. 

To expand on possible comorbidities, on rare occasions, TAPS may precede TTTS [30]. Some observations can be found with TAPS, such as placental discordance: the anemic donor has a thickened, hyperechoic placenta, and the plethoric recipient has a normal placental region, with clear demarcation between the two regions [14,24,44]. “Starry sky liver” is another ultrasound finding associated with TAPS, though more research is needed to validate. Starry sky appearance refers to a sonographic pattern of the liver, characterized by clearly identified portal venules (stars) and diminished parenchymal echogenicity (sky) that accentuates the portal venule walls [25]. Late-onset fetal growth discordance may also be observed (discordance of > 20% after 26 weeks of gestation) [13]. Late-stage TAPS symptoms, such as cardiac compromise, critically abnormal Doppler findings such as REDF, and hydrops, will eventually become noticeable on general MC ultrasound assessments [14]; however, waiting for such findings to appear increases fetal risk and compromises health. 

A 2016 literature review, which assessed 14 studies containing 29 pregnancies complicated by TAPS, showed that 48% of TAPS cases (*n* = 14) were spontaneous [45]. In another study of 179 monochorionic twins by Ashwal et al. [46], TAPS was diagnosed in 10 cases; eight of them were spontaneous, and two occurred following laser surgery. These findings reveal that spontaneous TAPS is a notable occurrence and, therefore, reliance cannot be placed on other complications accompanying TAPS. 

It is important to pause here and reflect on the idea of withholding routine MCA-PSV screening from patients until they begin to experience other MC complications. Not only are other MC complications not required to precede TAPS, but such belief trivializes the inherent risks that already accompany MC gestations. That is, they are all complicated.

In the influential 2010 paper, “There is NO diagnosis of twins,” Doctors Moise and Johnson reflect on the words of Dr. Kypros Nicolaides who said, “There is NO diagnosis of twins. There are only monochorionic twins or dichorionic twins. This diagnosis should be written in capital red letters across the top of the patient’s chart” [47]. Such a statement is born out of a profound understanding of the risks that accompany all MC twin pregnancies. Further, it seeks to instill a reverence that should accompany the decision-making process of those tasked with overseeing MC care. 

Nearly 10 years later, we find ourselves echoing Doctors Moise and Johnson. Within their paper, they argue the American Congress of Obstetrics and Gynecology needs to reevaluate and update their position regarding ultrasound frequency for MC twins, while creating a guideline that all MC twins receive bi-weekly ultrasounds beginning in gestational week 16 in an effort to provide timely detection of MC compilations such as TTTS and options for intervention. They contend, “Deferring an anatomy scan until 20 weeks of gestation once ‘twins’ have been diagnosed in the first trimester is no longer state-of-the-art in modern obstetrics” [47].

Likewise, deferring MCA-PSV screening for TAPS until other MC complications (possibly) present is no longer state-of-the-art in modern obstetrics. 

There is no rationale to support deferring TAPS screening until additional unsupported and unnecessary criteria have been met. Just as deferring routine bi-weekly ultrasounds and refusing to acknowledge inherent MC risks was detrimental to 2010 MC care, deferring routine TAPS screening and failure to acknowledge its inherent stand-alone risk is now detrimental to 2019 MC care. 

### 1.4. MYTH 4: There is an Unclear Treatment Protocol for TAPS

There are a host of effective treatments for TAPS, including fetoscopic laser surgery (the only causal treatment for TAPS [48]), intrauterine blood transfusion, intrauterine exchange transfusion, expectant management, elective preterm delivery, and selective feticide [14,26] each with accompanying benefits, drawbacks and considerations. 

A 2018 systematic review of morbidity and mortality associated with intrauterine interventions for the treatment of TAPS concluded that there is no mortality difference between any of the treatment modalities. Though expectant management may be associated with an increase in adverse perinatal outcomes when compared laser therapy and IUT [49].

In a retrospective study, where laser treatment for antenatally detected TAPS is compared to IUT or expectant management; laser therapy appeared to improve perinatal outcome by prolonging pregnancy and reducing respiratory distress syndrome [48]. The median time between diagnosis and birth was 11 weeks in the laser group compared to 5 weeks after intrauterine transfusion, and 8 weeks after expectant management. While laser therapy is the preferred treatment for TAPS, lack of TOPS can make the procedure challenging and puncturing the amniotic sac introduces risk [15].

Treatment with IUT in the donor can be performed either intravascularly or intraperitoneal. Intraperitoneal IUT is preferred, since intraperitoneal transfusion may allow slower absorption of red blood cells into the fetal circulation, preventing rapid loss of transfused blood in the circulation of the recipient twin [50]. It is important to note that IUT is not a causal treatment for TAPS; however it can provide temporary relief and allow for deferral of delivery, thus reducing the risks associated with severe prematurity [9]. Furthermore, potential negative side effects of IUT treatment include exacerbating polycythemia hyperviscosity syndrome in the recipient. A combination procedure of IUT in the donor and PET in the recipient can help reduce this risk [27]. 

With the first RCT underway, there is no current consensus on an optimal TAPS treatment protocol; to what extent individual interventions for TAPS improve fetal outcome is not fully understood; and what is deemed optimal will vary based on gestational age and other clinical observations. However, TAPS is associated with significant pre- and post-natal morbidity and mortality [9,26] and there is a current body of evidence to adequately guide providers and their patients until randomized controlled trials can be completed. We contend that the ultimate weight of treatment benefits and drawbacks rests with the patient and all current evidence should be provided to aid in decision making.

In the absence of evidence on optimal management, Tollenaar et al. [15] suggest that management decisions should be made after careful evaluation of different factors, including TAPS stage, gestational age, and the feasibility of the different types of intra-uterine intervention. They propose the following flow chart in Figure 1, which is not evidence-based at this time and represents expert opinion.

### 1.5. MYTH 5: If You Go Looking for Something to Be Wrong, You’re Going to Find It

If MCA-PSV Dopplers are being performed correctly, something that does not exist should not be found. As demonstrated in the discussion of Myth 2 (see Table 1), the sensitivity of MCA-PSV ≥ 1.5 MoM to predict severe anemia in TAPS donors has been shown to be 94% and specificity 74%. The sensitivity of MCA-PSV ≤ 1.0 MoM to predict polycythemia in TAPS recipients is even higher with a 97% and a specificity of 96% [24]. Additionally, diagnostic accuracy for delta MCA-PSV > 0.5 MoM for predicting TAPS shows a sensitivity of 83% and a specificity of 100% [20]. 

False positives can be further reduced by adhering to proper scanning technique, obtaining intra- and inter-rater consensuses and remaining mindful of additional conditions that may alter readings, such as previous transfusions. Importantly, performing serial MCA-PSV readings can reduce the false positive rate to only 5% [41].

There may be apprehension on behalf of some providers regarding MCA-PSV screening and its interpretation for diagnosing TAPS [12]; however, it should be accepted as a highly reliable test that can be reach maximum reliability with the implementation of proper training and technique. 

### 1.6. MYTH 6: MCA-PSV Doppler Screening Results will Just Give Pregnant Women Undue Stress

Women experiencing pregnancies with obstetric complications face increased rates of stress and anxiety [51,52] and MCDA twins are recognized for their high risk [47]. Rates of maternal anxiety typically decrease as the complicated pregnancy proceeds [53,54,55,56,57] however, Beauquier-Maccotta et al. [58] report that MC pregnancies not affected by TTTS demonstrated increased maternal anxiety as the pregnancy progressed. The percent of women with MC pregnancies not affected by TTTS meeting clinical cutoffs for anxiety increased from 20% when assessed at gestational week 20 to 29% by week 30. The authors suggest this may occur because the pregnancy carries elevated risk throughout the entire gestation. Per U.S. clinical guidelines, the MCDA population that does not demonstrate TTTS should already be receiving biweekly ultrasounds to screen for TTTS starting at week 16 until the conclusion of the pregnancy [11]. Although guidelines in England do not recommend MCA-PSV screening for uncomplicated MC pregnancies, they do recognize the utility of conducting multiple screenings during a single ultrasound session and fully informing patients about all testing [35]. Failure to perform routine MCA-PSV screening during an already intensive screening schedule based on concerns regarding maternal mental health ignores the reality that MCDA pregnancies are by their nature anxiety provoking and stressful experiences.

Additionally, steps can be taken to mitigate maternal anxiety with regards to screening and treatment for TAPS. Specifically, reducing parental uncertainty has been demonstrated to help alleviate maternal anxiety during medically complicated pregnancies [53,59]. This could include providing information about screening, diagnosis, and possible treatment plans.

### 1.7. MYTH 7: TAPS is Incredibly Rare and So Routine Screening is Not Necessary Unless the Patient is Post-Laser or Having Other Complications

The monozygotic twinning rate is roughly 3.5 per 1000 pregnancies, with approximately 70% being monochorionic [60], thus classifying MC twinning as “rare.” However, once a woman is confirmed monochorionic, her risks of TTTS, IUGR, congenital abnormalities, and TAPS are no longer rare; they are actually quite common. Up to 25% of monochorionic pregnancies are complicated by intrauterine growth restriction, TTTS, or unrelated intrauterine fetal death [61]. Up to 6% of MC twins will experience spontaneous TAPS [14] and up to 16% of those who had laser for TTTS will go on to experience TAPS [6]. We again caution that the figure particularly for spontaneous TAPS is likely underestimated due to low levels of routine ante and post-natal TAPS screening.

Nonetheless, six-percent represents a roughly 1 in 16 risk of being affected by a disorder that can result in serious fetal injury and/or death [27]. When considering the possible severity of the effects of TAPS, 1 in 16 becomes gravely serious and placing trust in “rarity” is no longer acceptable. 

We have already established that other complications preceding TAPS are not necessarily present (although not mutually exclusive either), emphasizing the need to routinely check for the presence of TAPS. 

### 1.8. MYTH 8: Routine MCA-PSV Doppler Screening Results in Too Many Unnecessary Premature Births

Risks associated with premature birth are well-established and should not be undervalued. This reality forces us to revisit the argument that TAPS diagnosis, and resulting conversations regarding treatment options, should always include a careful explanation of benefits, risks, and unknowns.

In the end, rarely will the best decision be abundantly clear. The primary argument of this paper, however, is that the decision ultimately rests with the patient. The patient must receive all of the evidence available—predicated upon MCA-PSV Doppler measurement values—so that she can manage her care and make informed decisions.

A 2016 literature review showed that amongst patients with spontaneous TAPS, average gestational age at birth was 32 weeks (23.6–38 weeks) [45]. When antenatal TAPS is detected, treatment options become available. Antenatal TAPS treatments can elongate a gestation. In a retrospective study where laser treatment for antenatally detected TAPS is compared to IUT or expectant management, the median time between diagnosis and birth was 11 weeks in the laser group compared to 5 weeks after intrauterine transfusion, and 8 weeks after expectant management [15]. Since diagnosis of TAPS at an earlier gestational age is associated with more favorable outcomes [62], routine MCA-PSV screening serves as a beneficial tool for minimizing TAPS-related morbidity.

## 2. Current U.S. Clinical Recommendations for TAPS Screening

As of 2019, the only current United States clinical recommendations regarding MCA PSV Doppler screening can be found in the Society for Maternal Fetal Medicine (SMFM) Publication Committee’s 2013 publication in the American Journal of Obstetrics and Gynecology, titled “Twin-twin Transfusion Syndrome” [11]. This recommendation paper, now over six years old, contains the following recommendation against the use of MCA-PSV Doppler to screen for twin anemia-polycythemia sequence (TAPS) at any time: “There is no evidence that monitoring for TAPS with MCA PSV Doppler at any time, including > 26 weeks, improves outcomes, so that this additional screening cannot be recommended at this time” [11]. 

The citation used to support this recommendation is based on the 2010 paper *Twin anemia-polycythemia sequence: diagnostic criteria, classification, perinatal management and outcome* by Slaghekke et al. [14]. However, use of this source to support SMFM’s guidance against MCA-PSV is perplexing given the Slaghekke et al. conclusion recommends routine MCA-PSV in all monochorionic pregnancies: “For timely detection and eventually treatment of TAPS cases, we recommend routine measurement of MCA-PSV with Doppler ultrasound on a regular basis (at least once every two weeks) in all monochorionic twins, in particular after laser treatment” [14]. 

Additionally, Table 4 in the 2010 Slaghekke et al. paper reveals that when TAPS was detected antenatally using MCA-PSV Doppler and interventions were performed (IUT, IUT with laser, and laser only), the result of those interventions was a 100% survival rate [14]. Again, this contradicts SMFM’s conclusion that there is no evidence monitoring with MCA-PSV improves outcomes [11].

While some fetal treatment centers have developed and disseminated their own screening standards, which include routine MCA-PSV screening starting at gestational week 16 (for example, see guidelines developed by The Fetal Center at Children’s Memorial Hermann Hospital) [30], our research demonstrates that several MFMs do not conduct these screenings for the reasons discussed above [12]. Given the SMFM publication is currently the only U.S. based clinical guidance on this topic, this single sentence recommendation provided by the SMFM publication to forgo testing has undue influence over the care offered and received by U.S. women pregnant with MC twins. 

## 3. Ethics of TAPS Screening and Management 

Because clinicians may have concerns regarding the validity of screening and diagnostic techniques as well as the risks associated with treatment, information related to MCA-PSV for routine TAPS screening is not consistently offered to patients. Providing patients with this information fits within a normative framework for reproductive health, involving three unified elements: (1) a dialectical, therapeutic relationship that balances the duties of providers and the rights of patients, (e.g., the duty of disclosure and right to information) and preserves the fiduciary role of clinicians; (2) considerations of justice that afford patients with fair and reasonable opportunities to secure their values and interests (e.g., parental rights), and; (3) ethical acts that promote the welfare of patients (beneficence), while avoiding unnecessary harms (nonmaleficence). Such frameworks that address challenges in prenatal screening and prenatal decision-making are presented in the literature [63,64,65]. The unified elements of this proposed framework guides a better understanding of reproductive freedoms and the dependent moral status of fetuses [66] while illustrating the prevailing ethical reasons for urgent policy and practice changes. These unified elements, as described below, are not discrete and should be considered holistically for optimal ethical guidance.

### 3.1. Preserving the Fiduciary Relationship

One of the most critical ethical considerations for TAPS screening is the development and preservation of the therapeutic, fiduciary relationship between clinician and patient, established through intimacy, trust, and care and requiring clinicians to prudently care for and act toward the benefit of their patients. In the context of routine TAPS screening, the therapeutic, fiduciary relationship requires clinicians to fully inform and disclose available screening options, recognizing the needs of patients as individual persons and not as statistics or probabilities. As part of their fiduciary responsibilities, clinicians are expected to learn new technologies and practices supported by evidence-based medicine (e.g., monitoring methods), implement current policies and recommendations into clinical practice, and address any concerns (i.e., TAPS screening) with colleagues and experts in the field. Fulfilling these obligations strengthens clinicians’ confidence and, more importantly, patient care. 

### 3.2. Justice and the Rights of the Pregnant Patient

Goals of justice require that patients are afforded fair and reasonable opportunities to secure their values and interests, including opportunities to achieve health regardless of clinicians’ preconceived beliefs of their interests and socioeconomic status. Non-directive counseling for routine TAPS screening is essential without clinician bias, criticism, or personal opinion. Withholding information is a violation of reproductive freedom and parental rights in securing the interests and the rights of their fetuses (i.e., a child’s right to an open future, which includes dependency rights) (See: Feinberg, 1980 [67]; Millum, 2014 [68]). By failing to fully inform a patient, the provider fails to acknowledge patient autonomy and reproductive rights, while inadvertently subjecting the patient to harms that could be properly managed, if not prevented entirely [69]. Withholding TAPS related information further restricts parental rights and decisions that taking into account the best interests of future children even if the best way to proceed is uncertain [70]. 

### 3.3. Beneficent Acts and Avoidance of Harm

The aforementioned evidence in this paper demonstrates that: (1) TAPS is a serious, potentially life-threatening complication, (2) treatment for antenatal TAPS is effective and includes either expectant management, IUT, or surgery, (3) and routine MCA-PSV, which has satisfactory sensitivity and specificity, is currently the only way to provide early detection of antenatal TAPS [65,71]. The minimal harms of the screening process itself (i.e., expectant management or surgery) significantly outweigh the potential harms of TAPS. Fully informing patients about the benefits and harms more broadly, including, but not limited to, potential financial, emotional, and social/familial burdens related to TAPS screening and outcomes, fulfills clinicians’ ethical duty of disclosure and promotes shared decision-making most consistent with patient values and judgments about what is best for their family [70]. 

Monochorionic twin pregnancies are inherently ethically complex pregnancies with the potential for difficult decisions [72], thus it is essential for clinicians, organizations, patients, families and others to be guided by the unified elements of an ethical framework, including the therapeutic, fiduciary relationship, justice and patient rights, and principles of beneficence and nonmaleficence. 

## 4. Conclusions

We recognize MCA-PSV screening for antenatal TAPS is currently not a standard of care for U.S. women pregnant with monochorionic twins. However, because of the information we present above, which not only dispels myths about MCA-PSV screening but also provides ethical arguments for screening, we present a call to action for U.S. organizations and clinicians to establish routine MCA-PSV screening as a standard of care for this high-risk population. We recognize the potential burdens this technology may pose for organizations and clinicians in terms of the time and effort it takes to establish fair guidelines and institutional policies and promote a standard of care unique to this population. However, the magnitude of life-altering burdens a patient may experience by not having the opportunity to be properly screened is much greater and can ultimately diminish the relationships patients have with their clinicians, and communities have with their health organizations. The patient and our future children require positive changes motivated by care and compassion rather than nocuous myths.

## Figures and Tables

**Figure 1 jcm-08-00977-f001:**
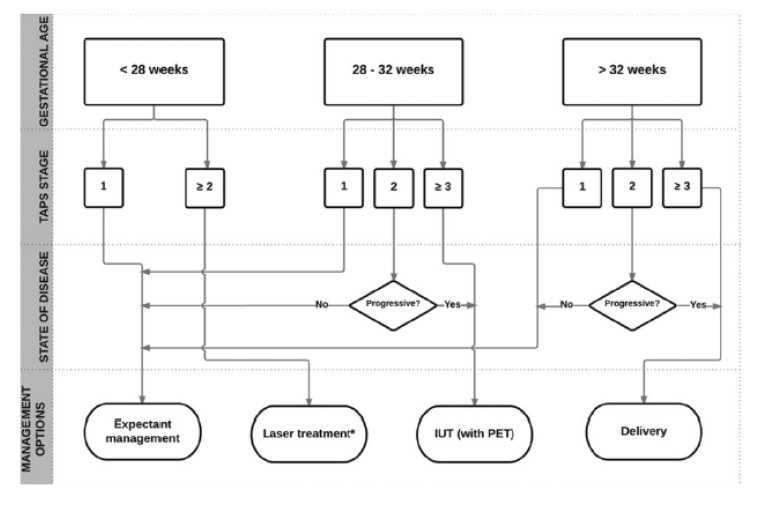
Flowchart with Antenatal Management Options for TAPS. Reproduced with permission from Tollenaar et al. Twin Research and Human Genetics; published by Cambridge University Press, 2016 [15].

**Table 1 jcm-08-00977-t001:** Accuracy of MCA-PSV MoM for Predicting Fetal Anemia and Polycythemia.

							Predictive Values	
Study	Sample	Interval between MCA-PSV Measure and Delivery/Reference Measurement	MCA-PSV Measurement	Outcome and Reference Standard	Sensitivity (95% CI)	Specificity (95% CI)	Positive (95% CI)	Negative (95% CI)
Slaghekke et al. 2015 [36]	43 TAPS pregnancies	Pre-natal MCA-PSV within 24 h of prenatal HB assessment or 24 hours of delivery	MCA-PSV ≥ 1.5 MoM	Severe anemia in TAPS donors as measured by prenatal or postnatal Hb levels	94% (85–98%)	74% (62–83%)	76% (65–85%)	94% (83–98%)
			MCA-PSV ≤ 1.0 MoM	Polycythemia in TAPS recipients as measured by prenatal or postnatal HB levels	97% (87–99%)	96% (89–99%)	93% (81–97)	99% (93–100%)
Tollenaar et al. 2019 [23]	35 TAPS pregnancies and 45 uncomplicated MC pregnancies	Pre-natal MCA-PSV within 1 week preceding delivery	MCA-PSV > 1.5 MoM in the donor; <1.0 MoM in the recipient	Postnatal TAPS	46% (30–62%)	100% (92–100%)	100% (81–100%)	70% (58–80%)
			Delta MCA-PSV > 0.5	Postnatal TAPS	83% (67–92%)	100% (92–100%)	100% (88–100%)	88% (77–94%)
Veujoz et al. 2015 [37]	40 TAPS pregnancies; 20 spontaneous; 20 post-laser	Pre-natal MCA-PSV within 48 h preceding delivery, or preceding in-utero transfusion	MCA-PSV > 1.5 MoM in the donor; <1.0 MoM in the recipient	Postnatal TAPS	71% (29–96%) *	50% (1–99%) *	83%	33%
					Area Under the Curve (AUC) (95% CI)			
Fishel-Bartel et al. 2016 [18]	69 MCDA pregnancies	Pre-natal MCA-PSV within 1 week preceding delivery	MCA-PSV >1.5 MoM in donor; <1.0 in MoM in the recipient	Postnatal TAPS	AUC = 0.87 (0.76–0.99)			

* Note: sensitivity and specificity confidence intervals were calculated and reported by NICE report. MCA-PSV = middle cerebral artery peak systolic velocity; HB/Hb = hemoglobin; MC = monochorionic; TAPS = twin anemia-polycythemia sequence; MoM = multiples of the median; MCDA = monochorionic-diamniotic; AUC = area under the curve; CI = confidence interval.

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
