# Peer review of "Dispelling Myths about Antenatal TAPS: A Call for Action for Routine MCA-PSV Doppler Screening in the United States"

_jcm, 2019, doi:10.3390/jcm8070977_

Round 1
Reviewer 1 Report
The authors are presenting a review of literature in support of screening for twin anemia polycythemia sequence starting at 16 weeks in monochorionic diamniotic twins. The manuscript is highly opinionated on certain aspects supported some of the studies. The article does not systematically address each questions with scientific rigor- a systematic review or meta-analysis of existing literature. Therefore, it is a very good letter to editor or an opinion, but lacks scientific merit to consider as a original article.
It does not address the concerns of the critiques of screening- the reference standards in the studies, weaknesses in the original studies, variability in definition of disease, and treatment effects on the TAPS.
Author Response
Thank you for your consideration of our manuscript, "Dispelling Myths about Antenatal TAPS: A Call for Action for Routine MCA-PSV Doppler Screening in the United States."
We appreciate the excellent suggestions for improving our manuscript. Please find
our responses to your comments in the table below.
Reviewer 1 | |
Reviewer Comment | Authors’ Response |
The authors are presenting a review of literature in support of screening for twin anemia polycythemia sequence starting at 16 weeks in monochorionic diamniotic twins. The manuscript is highly opinionated on certain aspects supported some of the studies. The article does not systematically address each questions with scientific rigor- a systematic review or meta-analysis of existing literature. Therefore, it is a very good letter to editor or an opinion, but lacks scientific merit to consider as a original article. It does not address the concerns of the critiques- the reference standards in the studies, weaknesses in the original studies, variability in definition of disease, and treatment effects on the TAPS. | In response to concerns about 1) the opinionated nature of the manuscript; and, 2) the lack of scientific rigor, we appreciate insights into ways in which the topic under examination could be developed into a systematic review or meta-analysis of existing literature. And, while such approaches may be a subsequent examination for our team or others tackling the gaps in clinical practice in regard to antenatal TAPS screening, it is essential that deeply embedded assumptions and practices in the U.S. are described in detail and critically examined from theoretical and pragmatic perspectives. That is, current gaps in antenatal screening processes are due, in part, to misinterpretation or misuse of existing literature, as well as, a lack of consideration for the significant progress in research and clinical practice that has been detailed in the selected literature that could yield a standard of care. Because clinical guidelines have been published that are counter to those valid studies we have identified in this manuscript, addressing this issue through a critical theoretical analysis is essential for prompting clinicians and readers to re-visit those guidelines through future research, i.e., the meta-analysis you have suggested. To start with a meta-analysis will likely impair the call to action, as such research may be exposed to the same fate as those TAPS projects that are underutilized or ignored. What we have provided is an argumentative review, which by definition, is a selective examination of literature to support or refute arguments, deeply embedded assumptions or philosophical problems, which, in this case, have led to guidelines and practices that negatively impact the welfare of our patient populations. This is our duty as researchers, clinicians, and ethicists. In terms of scientific rigor, the approaches to this paper follow the philosophical traditions of theoretical analysis, which is an accepted methodology in bioethics and the social sciences that intersect with medical research methods. The postulates, theoretical support, and argumentative framework substantiate our claims in which opinion is replaced with valid and sound logical argumentation and supported with foundational theoretical and descriptive justification from the medical literature. While the tone may be that of opinion, the descriptive and theoretical analysis, and concluding call to action, is consistent with the enterprise of research practices and collegial discussions. Thus, we respectfully believe this manuscript should be categorized as an original article, and not merely a letter to editor for which the gravity of this issue may go unnoticed.
We have added clarification of this issue within the manuscript.
Please see Page 2, Lines 66-72.
|
Reviewer 2 Report
Nicholas et al address in their review that routine MCA-PSV Doppler measurements should be performed to screen for antenatal Twin Anemia Polycythemia Sequence (TAPS). They high light and dispel the myths surround antenatal TAPS and the US guide line not to perform routine MCA-PSV Doppler measurements. This review is well structure and well written overview of the available evidence in the literature and recommends performing routine MCA-PSV Doppler measurements to screen for TAPS.
As addressed in the review potential risks for monochorionic twin pregnancies complicated by TAPS might be missed when routine screening for TAPS is not performed. TAPS can results in perinatal (severe) morbidity and mortality and therefore not screening for this condition can be potential harmful for TAPS patients and their families.
Some minor suggestions:
Page 3:
Line 85-98: Suggest to use the recently improved TAPS classification system and show stages based on the recent improved classification system by Tollenaar et al.
Line 99: other typical “ ultrasound” findings (suggestion to insert ultrasound)
Line 101: also starry sky liver in the recipient can be added as additional ultrasound finding
Line 116: suggestion to insert: permanently life altering for “TAPS survivors and their” families.
Page 6:
Line 185: suggestion to add: recommend to use the improved antenatal classification system by Tollenaar et al. Less TAPS cases will be missed and a delta is more representative for the difference within the twins.
Author Response
Thank you for your consideration of our manuscript, "Dispelling Myths about Antenatal TAPS: A Call for Action for Routine MCA-PSV Doppler Screening in the United States."
We appreciate the excellent suggestions for improving our manuscript. Please find our responses to your comments in the table below.
Reviewer 2 | |
Reviewer Comment | Authors’ Response |
Page 3: Line 85-98: Suggest to use the recently improved TAPS classification system and show stages based on the recent improved classification system by Tollenaar et al. | We have replaced our previous staging criteria example with an example of the improved classification system by Tollenaar et al.
Please see Page 3, Lines 102-111.
We chose to leave a brief explanation regarding the evolution of the use of MoM values to predict antenatal TAPS in an effort to assist readers who may not be aware of the recent and preferential delta standard.
|
Line 99: other typical “ultrasound” findings (suggestion to insert ultrasound) | Please see Page 3, Line 117.
We also deleted “antenatal” to limit repeated language in this portion of the paper.
|
Line 101: also starry sky liver in the recipient can be added as additional ultrasound finding
| Please see Page 3, Line 119. |
Line 116: suggestion to insert: permanently life altering for “TAPS survivors and their” families.
| Please see Page 4, Line 134. |
Page 6: Line 185: suggestion to add: recommend to use the improved antenatal classification system by Tollenaar et al. Less TAPS cases will be missed and a delta is more representative for the difference within the twins. | Please see Page 7, Lines 209-211. |
Reviewer 3 Report
This paper summarizes the value of routine performance of MCA-PSV measurements in monochorionic twins pregnancies in order to detect twin anemia polycythemia sequence (TAPS) during pregnancy. The authors aim to show evidence justifying routine MCA-PSV screening as a standard of care for patients with monochorionic twins. The issue of prenatal detection of TAPS discussed in this paper is important and the topic is well presented.
A few issues should be addressed:
1. When the authors report on the studies, which evaluated the accuracy of MCA-PSV in predicting TAPS, the interval between the MCA measurement and delivery should be mentioned for each study, as this interval may influence the performance of this measurement.
2. The authors should mention that value of MCA-PSV in predicting polycythemia by itself is not as well established as its use for prediction of fetal anemia. According to ref # 18, neonates with polycythemia had similar MCA-PSV measurements compared to neonates with normal hemoglobin level, and MCA-PSV>1MOM does not necessarily rule out fetal polycythemia.
3. In the section of MYTH 2 the authors should emphasize that it seems that Delta MCA-PSV rather the actual MCA-PSV of each twin is more accurate in prenatal detection of TAPS (ref 18 and ref 23).
4. In the section of MYTH 4 the authors should emphasize that the different interventions for TAPS haven’t been shown to improve perinatal outcome and that the flow chart presented is an expert opinion and not evidence-based.
5. In the section of MYTH 4- in the paragraph discussing IUT treatment, the authors should add that IUT does not only provide temporary relief, but allows deferral of delivery and thus reduces the risk associated with severe prematurity.
Author Response
Thank you for your consideration of our manuscript, "Dispelling Myths about Antenatal TAPS: A Call for Action for Routine MCA-PSV Doppler Screening in the United States."
We appreciate the excellent suggestions for improving our manuscript. Please find our responses to your comments in the table below.
Reviewer 3 | |
Reviewer Comment | Authors’ Response |
When the authors report on the studies, which evaluated the accuracy of MCA-PSV in predicting TAPS, the interval between the MCA measurement and delivery should be mentioned for each study, as this interval may influence the performance of this measurement.
| For the four studies included in Table 1 we have added a column to include information related to the interval between the timing of prenatal MCA-PSV and delivery.
Please see modified Table 1 on Pages 5-6 (yellow highlights).
We have also added in the following sentence to the manuscript, “The four studies in Table 1 included in their protocols a timely interval between prenatal MCA-PSV and delivery (no greater than 1 week) increasing confidence in the accuracy of the MCA-PSV measurements.”
Please see Page 4, Lines 164-166.
|
The authors should mention that value of MCA-PSV in predicting polycythemia by itself is not as well established as its use for prediction of fetal anemia. According to ref # 18, neonates with polycythemia had similar MCA-PSV measurements compared to neonates with normal hemoglobin level, and MCA-PSV>1MOM does not necessarily rule out fetal polycythemia.
| Please see Page 4, Lines 153-155. |
In the section of MYTH 2 the authors should emphasize that it seems that Delta MCA-PSV rather the actual MCA-PSV of each twin is more accurate in prenatal detection of TAPS (ref 18 and ref 23).
| We have taken steps to clarify and emphasize this point in numerous parts of the manuscript, including under MYTH 2.
Please see, Page 3, Lines 98-99
Please see Page 3, Lines 102-111.
Please see Page 5, Lines 182-183.
|
In the section of MYTH 4 the authors should emphasize that the different interventions for TAPS haven’t been shown to improve perinatal outcome and that the flow chart presented is an expert opinion and not evidence-based
| Please see Page 9, Line 299.
Please see Page 9, Lines 309-310.
Additionally, we have added a brief statement to this portion to help emphasize the premise of the paper.
Please see Page 9, Lines 303-305.
|
In the section of MYTH 4- in the paragraph discussing IUT treatment, the authors should add that IUT does not only provide temporary relief, but allows deferral of delivery and thus reduces the risk associated with severe prematurity.
| Please see Page 9, Lines 293-294. |
Round 2
Reviewer 1 Report
The authors are making a case for argumentative and ethical consideration for the study. However, the scientific question of whether screening TAPS and implications of false positive or false negatives are not explored.
Author Response
Reviewer 1 | |
Reviewer Comment | Authors’ Response |
The authors are making a case for argumentative and ethical consideration for the study. However, the scientific question of whether screening TAPS and implications of false positive or false negatives are not explored. | The issue of how Doppler MCA-PSV false positives and/or false negatives influence TAPS screening is explored in MYTH 2 and MYTH 5.
We address this issue in a few different ways:
1) By discussing the specific levels of sensitivity (used to assess false positives) and specificity (used to assess false negatives) as outlined in the relevant literature.
These findings can be found in Table 1 (pages 5-6) as well as on pages 4-5, Lines 169-181.
We have added additional clarification on Page 7 to include the specificity/false negative findings. Please see Page 7, Line 207.
Overall findings confirm that MCA-PSV Doppler sensitivity and specificity levels are high and acceptable.
2) By reviewing the literature to suggest ways in which MCA-PSV Doppler screening technique can be adjusted to maximize accurate screening results.
Please see Page 4, Lines 144-151.
Additionally, we include discussion regarding the value of adopting of delta MoM values to catch more TAPS cases and the use of serial MCA-PSV screening to reduce false negatives.
Please see Page 7, Lines 196-212.
This is reiterated on Page 10, Lines 316-328.
3) By discussing the ability for MCA-PSV Doppler screening results to prompt interventions that may result in premature birth.
Please see MYTH 8 on Page 11, Lines 368-382.
4) By discussing clinicians’ prevailing ethical obligation and rights of the patient to be offered the option for TAPS screening regardless of the (low and acceptable) possibility of false positives and/or false negatives.
Please see Page 12, Lines 419-427.
Finally, we would like to highlight that the discussion of the implications of faulty MCA-PSV Doppler readings is an important but minor portion of the overall paper. That is, there are many prevailing myths working to withhold routine TAPS screening from women in the United Sates and issues surrounding potential false positives and/or false negatives represents a small portion of these myths. Therefore, false positives and/or false negatives are not the main focus of the paper, and are thus discussed in proportion to the influence they are having on the withholding of routine TAPS screening.
Complementary to this, the entire paper is founded on the argument that despite inherent flaws in testing or established treatment protocol (which throughout the paper we admit exist), the only person who can make the decision to accept or reject the risks and benefits of routine TAPS screening is the patient herself. |